# Mesocorticolimbic and Cardiometabolic Diseases—Two Faces of the Same Coin?

**DOI:** 10.3390/ijms25179682

**Published:** 2024-09-06

**Authors:** Csaba Papp, Angela Mikaczo, Janos Szabo, Csaba E. More, Gabor Viczjan, Rudolf Gesztelyi, Judit Zsuga

**Affiliations:** 1Department of Psychiatry, Faculty of Medicine, University of Debrecen, Nagyerdei krt. 98, H-4032 Debrecen, Hungary; dr.papp.csaba@gmail.com (C.P.); szaboj@med.unideb.hu (J.S.); dr.more.csaba@kenezy.unideb.hu (C.E.M.); zsuga.judit@med.unideb.hu (J.Z.); 2Department of Pulmonology, Faculty of Medicine, University of Debrecen, Nagyerdei krt. 98, H-4032 Debrecen, Hungary; angelamikaczo@gmail.com; 3Doctoral School of Pharmaceutical Sciences, University of Debrecen, Nagyerdei krt. 98, H-4032 Debrecen, Hungary; 4Department of Pharmacology and Pharmacotherapy, Faculty of Medicine, University of Debrecen, Nagyerdei krt. 98, H-4032 Debrecen, Hungary; viczjan.gabor@pharm.unideb.hu

**Keywords:** mesocorticolimbic system, noncommunicable diseases, machine learning, target prioritization, network analysis

## Abstract

The risk behaviors underlying the most prevalent chronic noncommunicable diseases (NCDs) encompass alcohol misuse, unhealthy diets, smoking and sedentary lifestyle behaviors. These are all linked to the altered function of the mesocorticolimbic (MCL) system. As the mesocorticolimbic circuit is central to the reward pathway and is involved in risk behaviors and mental disorders, we set out to test the hypothesis that these pathologies may be approached therapeutically as a group. To address these questions, the identification of novel targets by exploiting knowledge-based, network-based and disease similarity algorithms in two major Thomson Reuters databases (MetaBase™, a database of manually annotated protein interactions and biological pathways, and Integrity^SM^, a unique knowledge solution integrating biological, chemical and pharmacological data) was performed. Each approach scored proteins from a particular approach-specific standpoint, followed by integration of the scores by machine learning techniques yielding an integrated score for final target prioritization. Machine learning identified characteristic patterns of the already known targets (control targets) with high accuracy (area under curve of the receiver operator curve was ~93%). The analysis resulted in a prioritized list of 250 targets for MCL disorders, many of which are well established targets for the mesocorticolimbic circuit e.g., dopamine receptors, monoamino oxidases and serotonin receptors, whereas emerging targets included DPP4, PPARG, NOS1, ACE, ARB1, CREB1, POMC and diverse voltage-gated Ca^2+^ channels. Our findings support the hypothesis that disorders involving the mesocorticolimbic circuit may share key molecular pathology aspects and may be causally linked to NCDs, yielding novel targets for drug repurposing and personalized medicine.

## 1. Introduction

According to the United Nation’s (UN’s) “Sustainable Development Goals set forth in 2015”, all countries are committed to reducing premature deaths due to noncommunicable diseases (NCDs) by 33.3% by 2030. Despite this commitment, according to the 2022 Progress Monitor, only a few countries are on their way to meet these objectives [1]. Lagging results are incomprehensible, as evidence-based guidelines are in order for both preventing and managing NCDs [2]. The controversy may be resolved if the nature of NCDs is taken into consideration, as the most common risk factors for the four most prevalent noncommunicable diseases e.g., cardiovascular diseases, cancer, diabetes and chronic respiratory diseases are alcohol misuse, unhealthy diets, smoking and sedentary lifestyles. Given these antecedents of NCDs, the responsibility of the individual should be considered, especially in light of the fact that these four risk behaviors are all linked to the mesocorticolimbic (MCL) system (for a brief overview, see [3,4]). Hence, it may be proposed that altered reward-related processes contribute to the development of risk behaviors paving the path to NCDs. The ongoing discourse to include common mental disorders (CMDs) under the umbrella term of NCDs must also be noted. Shared etiology, high co-morbidity and overlapping pathways between NCDs and CMDs has been established previously [5].

Alteration of MCL function is well established in the case of the harmful use of alcohol [6], nicotine addiction [7] and hedonic eating [8]. Additional to the aforementioned risk behaviors, numerous mental disorders, e.g., schizophrenia [9], depression [10], anxiety [11], eating disorders [12], pathological gambling [13], post-traumatic stress disorder [14], have also been associated with altered MCL function. Thus, it may be proposed that alteration of the MCL system may be a common factor in the etiopathogenesis of several risk behaviors and mental disorders and thus may emerge as a transdiagnostic marker linking NDCs and CMDs. Furthermore, by identifying shared molecular pathways, novel molecular targets that are potentially exploitable for drug repositioning may be identified. The need for new therapeutic approaches is pressing given the unmet medical need for combating NDCs and CMDs. A possible way to bridge the gap for unmet medical needs is to identify novel targets for drug repurposing. With increasing barriers for innovative de novo drug development (e.g., financial, regulatory and time constrains), focus has shifted toward identifying new targets for existing drugs. Identification of new targets may lead the way to target-, knowledge- and network-based drug repositioning [15,16].

Better understanding and identification of the common neurobiological underpinnings of diverse symptoms may yield new therapeutic targets. By unveiling novel therapeutic targets, personalized approaches may be identified, leading to higher response rates in therapy [17]. Although classified under distinct diagnostic categories, altered functioning of the MCL system has been described for various mental disorders as well as risk behaviors (see above). Thus, we set out to test the hypothesis that the mesocorticolimbic system is a common denominator contributing to the pathogenesis of diverse mental disorders and risk behaviors and as such may be approached therapeutically as a group.

## 2. Results

### 2.1. Performance of the Model

The performance of the model in terms of prediction error as a function of the number of features included in the model is shown in Figure 1. Removing features increased model performance up to a point (e.g., 47 features), because removing the less informative features reduced noise and biases. However, as the removal process progressed, elimination of explanatory features led to information loss and poorer predictive model performance.

The performance of the final model, as assessed by the area under curve for the receiver operator curve was 0.927, reflecting a good accuracy for the model, with good ability to recover a large portion of what is already known (Figure 2). The variation of model performance was ~90% when assessed in the cross-validation process over ten repeats of the outer loop (Figure 3).

### 2.2. Targets for Treating MCL Disorders

The integrative prioritization of proteins as candidate targets by machine learning yielded several potentially new therapeutic targets. Among the highest ranked 250 targets, 73 (53%) of the 137 predefined positive control genes/targets were recovered, corresponding to an approximate known target enrichment of 36.6-fold over baseline. (For the list of the top 250 targets yielded by machine learning see Appendix A). The average rank score in the set of the top 250 targets was 188.4 (median 189.3), with a minimum of 4.2 and a maximum of 339.4. The worst score in the entire table was 16,589 for the considered total of 18,493 genes.

Using information curated in MetaBase^TM^, the existence of direct connections between the highest ranked 250 targets was assessed. Utilizing the information from literature-derived protein–protein interaction targets seemed to closely interact, representing a densely interconnected system involved in neurotransmission and related functions of the central nervous system (Figure 4).

### 2.3. Assessment of the Top 25 Control and Novel Targets

The top 25 control target genes yielded by the final integrated model are shown in Table 1. The highest ranked target in the control target set was dopamine receptor D2. It should be noted that all dopamine receptors scored highly in the feature matrix, though not equally so: DRD2 (rank 1), DRD3 (rank 9), DRD4 (rank 19), DRD1 (rank 45), DRD5 (rank 278). The current results point to the higher relevance for the D2-like family (D2, D3, D4) regarding MCL disorders than the D1-like family (D1, D5). Another example of a well-established high scoring target is monoamine oxidase B (MAO-B). This target attained the second highest predicted score.

Upon further assessment of the enriched molecular functions of these top 25 control targets, it may be observed that these targets are mainly involved in neurotransmission and modulation of membrane potential (Table 2).

Taken together, the analysis recovered a number of previously well-established targets for disorders that belong to the group of MCL disorders at the top of the prioritization. This suggests that this computational model successfully captured current knowledge and thus may be used to generate novel predictions regarding putative targets for drug development in the field of MCL disorders.

### 2.4. Putative Targets for MCL Disorders

Assessment of the top 25 novel targets retrieved by the model yielded serendipitous results (Table 3).

Highly ranked novel targets included DPP4, PPAR-gamma, NOS−1, BDNF, IL1 and IL6 (see Appendix A). The highest ranked novel target in this analysis was dipeptidyl-peptidase 4 (DDP4), a ubiquitous, intrinsic membrane glycoprotein. This target is mainly exploited for the management of hyperlipidemia, hypertension and diabetes. Another highly ranked novel target was peroxisome proliferator-activated receptor gamma. Similarly to DDP4, PPAR-gamma is mainly used in the context of metabolic diseases but not in psychiatry. Components of the renin-angiotensin aldosterone system, e.g., angiotensin converting enzyme (ACE) and angiotensin II receptor type 1, were also highly ranked, molecular targets mainly associated with the function of the cardiovascular system and metabolism. Furthermore, it should be noted that a key neurogenesis factor, brain derived neurotrophic factor (BDNF), was also predicted to be tightly connected with MCL disorders. Interestingly, IL1B and IL6 also scored relatively highly among the novel targets (55th and 69th out of thousands of proteins in the prioritization). When further evaluating the top 10 gene ontologies of molecular function with enrichment in this group of targets, components of Ca^2+^ signaling converging on cAMP response element binding protein 1 (CREB−1) emerged (Table 4).

## 3. Discussion

The findings of the current study support the hypothesis that disorders involving the mesocorticolimbic circuit may share key molecular pathology aspects. Furthermore, this study aimed to identify the common targets for mesocorticolimbic system disorders as a group and test a hypothesis that these disorders may be approached therapeutically as a group. Selection of the initial scope of disorders was based on prior knowledge regarding the involvement of the mesocorticolimbic circuit. Using a combination of approaches, e.g., knowledge-, network- and disease similarity-based methods, candidate targets were retrieved for prioritization, leveraging the Thomson Reuters’ bioinformatics portfolio. Each approach characterized proteins from a specific perspective to assess the likelihood of a protein to represent a target for the MCL disorder as a group. Next, using machine learning the individual approaches were combined into an integrated score, resulting in a final target prioritization. Machine learning identified characteristic patterns of the already known targets (control targets) from the final model, with high accuracy (the AUC of the ROC curve was ~93%). This is similar to model performance reported by others [18]. As machine learning analysis was trained on the union of targets for MCL disorders, this high-accuracy coherent prediction, with the majority of already established targets readily being recovered at the top, further underscores the hypothesis of joint etiology. This result is unlikely for a collection of disconnected disorders, suggesting that these pathologies share key molecular links. Next, using rules obtained on this training set all other proteins (~18,000) were prioritized. The analysis resulted in a prioritized list of 250 targets for MCL disorders, many of these targets were already known to be involved with the mesocorticolimbic circuit. Upon construction of the molecular network from the highest ranked 250 targets, a densely interconnected network emerged, further supporting the hypothesis that the molecular targets of MCL disorders represent a molecular system with numerous avenues for crosstalk. The well-established top targets included dopamine receptors, MAOs and serotonin receptors (Table 1), whereas novel (not part of the training set) targets included DPP4, PPARG, NOS1, ACE, ARB, CREB1, proopiomelanocortin and diverse voltage-gated Ca^2+^ channels (Table 3). These targets may represent candidates for further investigation in the examined disorders.

One of the fundamental characteristics of the MCL circuit is the involvement of the dopamine pathway (for an overview, see [3,4,19]). Dopamine receptors are primarily located in the caudate, putamen, nucleus accumbens and olfactory tubercle, where they are involved in the modulation of locomotion, reward, reinforcement, memory and learning [20,21]. Thus, the mesocorticolimbic system may be regarded as a dopamine system, and an entire field of study concerns itself with the critical importance of dopamine signaling in the brain. In the current study, every dopamine receptor received a high rank, although the D2-like family (D2, D3, D4) seemed slightly more relevant for the treatment of MCL disorders than the D1-like family (D1, D5). Another example of a well-established highly ranked target is MAO-B (this target received the second highest rank). MAO-B is one of the two MAO isoforms that catalyze the oxidation of monoamines including dopamine, serotonin and adrenalin. Dopamine is mainly metabolized by MAO-B. Abnormal regulation of MAO-B is associated with depression, substance abuse and attention deficit disorder, among others [22]. Thus, it seems that when MCL disorders were considered as a group, some generally known molecular targets were retrieved by the model.

That machine learning was able to predict known drug targets for a seemingly diverse group of mental disorders with high accuracy suggests that the computational model captures existing knowledge well. Taking these concepts one step further, the ability of this approach to make predictions regarding novel, unknown disease-specific drug target candidates for future research could be proposed. Others have shown that diseases that share a high number of differentially expressed genes have similar underlying pathologies and pathways [18]. Using the results of target-based clustering, Emig and colleagues were able to identify diverse diseases that share common underlying biological processes and, based on these findings, novel disease-specific drug target candidates emerged [18].

It is interesting to note that several known targets for metabolic diseases, e.g., diabetes and insulin resistance, emerged from the model prediction. The highest ranked novel target, DDP-4, is a ubiquitous, intrinsic membrane glycoprotein and a serine exopeptidase that assumes a role in the post-translational modification of membrane proteins by cleaving X-proline dipeptides from the N-terminus of polypeptides. DDP-4 is present on the luminal side of epithelial and endothelial cells and on monocytes, lymphocytes and natural killer cells. The key enzymatic function of DDP-4 is the degradation of incretins, a polypeptide that stimulates pancreatic α and β cells and regulates glucose-stimulated insulin secretion [23]. Membrane-bound DDP-4 may be cleaved and its enzymatically active soluble form is present in the circulation and was implicated in playing a causative role in development of insulin resistance [24]. The third highest candidate target was PPAR-γ, which is known to improve insulin sensitivity and glucose metabolism by promoting fatty acid uptake and triglyceride formation and storage in lipid droplets [25]. Furthermore, several other targets with links to insulin sensitivity were ranked highly among the list of novel targets, like angiotensin converting enzyme and angiotensin II receptor type 1 [26]. Neuronal nitric oxide synthase, additional to playing a role in insulin sensitivity, assumes an important role in the central regulation of glucose homeostasis, in fact the central actions of apelin, ghrelin and leptin are known to be nitric-oxide-dependent processes (for an overview, see [27]). The emergence of druggable targets classically related to glucose and lipid homeostasis in the context of MCL disorders is worthy of notice. That hunger is a primary driver for organisms is well established [28]. That sweet taste has inherent hedonic value and is considered as a primary reinforcer is also well known [29]. Starting from that motivation, incentive salience and reward processing are at the heart of the MCL system; the emergence of novel targets that are involved in metabolism point to the fundamental two-way interaction between somatic and mental function. In agreement with these findings, others report similar results. In a small community sample of 109 healthy adults with normal blood pressure (120–139/80–89 mmHg), fronto-striatal cerebral blood flow was assessed in response to cognitive challenge and cardiometabolic risk was assessed at follow-up after 2 years. In this prospective study, the authors found that greater activation of fronto-striatal brain regions was associated with increases in cardiometabolic risk factors at follow-up [30], suggesting an interplay between the MCL circuit and the cardiovascular system. Heart–brain interactions in the context of mental disorders, e.g., dementia, depression, as well as vascular diseases (ischemic heart disease, heart failure stroke and Takotsubo syndrome), have been reviewed previously. The authors articulated the central role of the limbic system as forming the link between the two organ systems [31]. Furthermore, they emphasized the need to further investigate the molecular mechanisms regulating the brain–heart interaction with the hope of identifying new individualized treatments that interrupt the pathogenic crosstalk.

Our findings that the MCL system may be a common pathogenic factor underneath CMDs and NCDs are further corroborated by epidemiologic observations. Comorbidity between cardiometabolic diseases and mental disorders has been reported in a large cross-sectional study based on the UK Biobank. Overall, data from 391 083 participants with available information regarding cardiometabolic diseases (CMDs) and depression was analyzed. The presence of CMDs showed a dose-dependent and accumulative relationship with an increased risk of depression disorder after multivariable adjustments [32]. Conversely, an increased risk for cardiovascular disease was identified in a longitudinal cohort study. The cohort included 900,240 patients newly diagnosed with psychiatric disorders, their 1,002,888 unaffected siblings and 1:10 sex- and age-matched reference population with no prior diagnosis of cardiovascular diseases at enrollment. The follow-up period was 30 years. A direct link between psychiatric disorders, hypertension, ischemic heart disease, angina pectoris, venous thromboembolism and stroke was established independent of familial factors. The authors concluded that people with psychiatric disorder are at increased risk for CVDs and that increased surveillance and treatment of CVDs should be an integral part of their clinical management [33]. Moreover, the comorbidity of several cardiometabolic diseases and mental disorders has also been described in smaller studies [34], e.g., insulin resistance and ADHD [35], heart failure and cognitive impairment [36], and myocardial infarct and anxiety [37].

The study has some limitations in terms of its potential for translation, as data-driven identification of novel therapeutic drug targets may not yield viable therapeutic options due to, for example, dosing issues (e.g., s higher dose is needed for the new indication, raising concerns regarding safety), considerable off-target effects and possible barriers coming from intellectual property claims [38]. Furthermore, the current analysis is based on two proprietary databases not in the public domain, which limits independent access. However, this adds some additional merit to the study, by allowing drawing inferences from such databases. A further limitation of the study is the need for future clinical validation. However, given the fact that several currently marketed drugs are available for several of the drug targets newly identified within the current study, new candidates with well-established safety profiles may be considered for proof-of-concept trials. For example, a PPARG agonist (e.g., pioglitazone), DDP4 antagonists (e.g., sitagliptin, linagliptin, etc.), ACE inhibitors (enalapril, captopril, etc.) and AGTR1 blockers (e.g., telmisartan, losartan, etc.) are currently available for various cardiometabolic indications, thus may serve as feasible candidates to pursue clinical proof-of-concept studies for use in MCL disorders. Alternatively, retrospective analysis of clinical records or health claim records may provide preliminary evidence for the translation potential of currently used medications (for example the use of ACE inhibitors for cardiomyopathy in patients suffering from comorbid depression).

In conclusion, our findings support the hypothesis that disorders involving the mesocorticolimbic circuit share key molecular pathways and may be causally linked to NCDs. Looking at the leading NCDs and their risk factors paralleled by the co-existence of CMDs, it is plausible that the intertwining relations between known targets for the MCL circuit and cardiometabolic diseases may yield already known well-tolerated drugs for drug repurposing and personalized medicine.

## 4. Materials and Methods

### 4.1. Selection of Disorders and Positive Control Targets

The current analysis included mental disorders with prior knowledge regarding the mesocorticolimbic system’s involvement in their etiopathogenesis. Disorders included in the Thomson Reuters databases Integrity^SM^ and Metabase^TM^ were included in the analysis. Accordingly, Alzheimer’s disease, anorexia nervosa, anxiety, bipolar disorder, bulimia, depression, mania, obsessive-compulsive disorder, Parkinson’s disease, pathological gambling, phobic disorders, post-traumatic stress disorder and schizophrenia were selected for the current study (from here on collectively referred to as MCL disorders). After defining the scope of the mental disorders’ network- and pathway-based algorithms, knowledge and statistical inference were used to identify links between a disorder and its target protein(s).

Integrity^SM^ (http://integrity.thomson-pharma.com (accessed on 6 February 2016), a unique knowledge solution integrating biological, chemical and pharmacological data, access in not in the public domain) is a collection of integrated biological, chemical and pharmacological data on 400,000 compounds and 220,000 patient family records. Molecular targets, indication of use and clinical phase of development are assigned to each drug. According to the clinical development phase, drug targets may be validated, candidate, exploratory or none. The drug targets associated with launched drugs or drugs under active development for the indication of the disorder of interest are considered validated targets, and targets associated with drugs that are no longer under active development for the respective disorder are considered candidate targets. Exploratory drug targets are related to drugs that are under biological investigation for the disorder. Drug targets not assigned any status were not included in the current analysis. For each disorder, Entrez Gene drug target associations were identified and served as true positives for each disorder for evaluating drug target predictions. Targets that were associated with at least two of the predefined MCL disorders were used as positive controls when model predictions were assessed. Overall, 137 common druggable targets were identified based on information coded in the Integrity^SM^ database.

MetaBase™ (a comprehensive, manually curated and semantically consistent knowledge source, access is not in the public domain [39]), on the other hand, is a database of manually annotated protein interactions and biological pathways. This database stores each molecule as a network object and contains the functional description of the molecule (receptor, kinase, transcription factor, etc.). Network objects may be proteins, protein complexes or protein families, or small molecules including non-coding RNAs and metabolites. The interactions included in MetaBase™ have been manually compiled from the literature and represent a physical relationship between pairs of elements in the network. The interactions are marked with additional information regarding bidirectionality (activation, inhibition or unspecified) and mechanism of action (e.g., binding, phosphorylation). Only protein–protein interactions with direct mechanisms were used in this study. Accordingly, the mechanisms included: binding, competition, transformation, cleavage, catalysis, transport, receptor binding, transport catalysis and covalent modification (phosphorylation, dephosphorylation, ubiquitination, deubiquitination, sumoylation, desumoylation, neddylation, deneddylation, etc.). Indirect mechanisms (e.g., influence on expression) were omitted.

In addition, MetaBase™ stores information on linear pathways, representing the signaling cascade from receptor activation to the cellular response and pathway maps representing pathway depictions in a particular biological context. Pathway maps include several linear pathway components with crosstalk and regulatory feedback loops among them. Regular pathway maps capture normal signaling pathways. Disease-specific pathway maps depict signaling events most relevant to a disease or disorder in focus, e.g., gain or loss of protein function resulting in new or disrupted protein interactions compared with the respective normal pathway, etc. In the current study, over 700 regular and over 600 disease pathway maps were used from the MetaBase™ portfolio.

To attest if a common core molecular network could be identified for mental disorders with known links to the MCL system, proteins as targets were prioritized in this respect. Knowledge-based, network-based and disease similarity-based approaches were used to retrieve potential targets. The different approaches produced various scores for each protein encoded in the human genome. The scores were integrated into a final score by machine learning analysis in a way where integration combined the different scores with appropriate weights. Weights were iterated using the weights from targets that are already known for the disorders in question.

### 4.2. Knowledge-Based Approaches

Using knowledge-based approaches: a protein’s role in MCL disorders was assessed based on prior knowledge in terms of this protein’s connection to pathways, molecular interactions and biomarkers involved in the disorder. Prior knowledge was defined as information available in the Integrity^SM^ and MetaBase^TM^ databases. Overall, five knowledge-based methods were used as follows. First, the established biomarkers for each mental disorder involved in the analysis were inferred jointly from the two databases. For each indication, proteins received a score of 1 or 0 for being a biomarker or not, respectively. Next, using the MetaBase^TM^ network, interaction inference was assessed by identifying the targets (proteins) that interact with targets known to be involved in the disorder. All 1-step-away direct neighbors of these proteins were identified and tested for enrichment with known biomarkers of each disorder under investigation. A score of a log-transformed enrichment *p*-value was computed for each protein. Third, using the MetaBase^TM^ pathway maps, enriched pathway inference was retrieved to predict novel targets. Signaling pathways significantly enriched in biomarkers of the disorders were identified and proteins in these pathways were predicted as candidate therapeutic targets. Computationally, each pathway was tested for an enrichment in disease biomarkers yielding a −log(*p*-value) associated with each pathway in the database. Next, these pathway scores were transformed into protein scores by adding each pathway score for a protein of interest across relevant pathways (e.g., where a protein of interest is present). The fourth approach used pathways specifically constructed for biological processes or diseases curated by the MetaMiner consortium. For the present analysis, the pathways reconstructed for biological areas related to central nervous system diseases and disorders were assessed (e.g., “Neurophysiological process”, “Depressive Disorder”, “Mental Disorders”, “Parkinson Disease”; as well as general biological themes informing us on basic protein functions in the cells: “Carbohydrates metabolism”, “Energy metabolism”, “Energy metabolism”, etc.). The score for a potential target was obtained by counting the number of pathways on which the protein of interest was present. Finally, linear pathway inference was performed to retrieve potentially new targets using MetaBase™. In linear pathway inference, a protein is predicted as a target if it is frequently found on linear pathways containing biomarkers of the disease. For the present analysis, the inclusion criteria for linear pathways was to start with a ligand–membrane receptor interaction and end with a transcription factor. A protein score was calculated as the ratio of the number of linear pathways containing the protein of interest together with at least one known biomarker for the disorders divided by the total number of linear pathways containing the protein of interest.

### 4.3. Network-Based Approaches

Network-based approaches were used to predict novel protein–disorder associations based on the topological positions of proteins in the interactome. Network algorithms were used to assess the probability of a protein acting as a disease driver by testing its topological significance in the MetaBase™ network with regard to known biomarkers of the disorders. Four network algorithms were used to capture the complementary topological properties of the targets: network propagation, random walk, interconnectivity and neighborhood scoring. These methods score proteins in the MetaBase™ network by how strongly they are topologically associated with the biomarkers of the input disorders.

Network propagation, a global method, was performed as described previously [40]. The starting nodes for network propagation were the predefined biomarkers for MCL disorders, from which flow is injected into the network. In each iteration, the flow was diffused further through the network until the changes of flow reached equilibrium. The final flow level for each node described the overall proximity of the node to all the starting points. To compensate for bias due to the network structure, node degrees were taken into account during computations and edge weights were normalized to the node degrees of the adjacent nodes. In each iteration t, the flow for each node was computed as follows:$$F_{t}=\alpha \cdot W^{\prime }\cdot F_{t-1}+\left(1-\alpha \right)\cdot Y$$

F is a vector containing the flow in iteration t. α is a weighting factor, set to 0.8 in this study. W’ is the transition matrix (normalized by the node degrees). Y is the prior knowledge vector that contains a 1 for each starting node and 0 for all other nodes.

Random walk, a global method that utilizes the topology of the whole network, is defined as the transition of an iterative walker from its current node to a randomly selected neighbor [41]. The starting points for the random walkers were the known biomarkers for MCL disorders, with random walkers moving to a neighboring node with equal probability at each iteration. Random walkers may also return to their starting points (random walk with restart). At each iteration the nodes are assigned a score that reflects the probability that a random walker visits the node at some point. Once the probability scores reach equilibrium, the random walk terminates. In each iteration t, the probability scores for the nodes are computed as follows:$$p^{t}=\left(1-r\right)\cdot W\cdot p^{t-1}+r\cdot p^{0}$$

p is a vector containing the node probabilities in iteration t. r is a weighting factor describing the restart probability, set to 0.75 in this study. W is the transition matrix that is normalized to fulfill the properties of a stochastic matrix. p^0^ is the starting vector that contains equal probabilities for each starting node and 0 for all other nodes.

Interconnectivity is a local method for prioritizing targets based on their overall connectivity to the set of starting points (e.g., biomarkers) [42]. The method accounts for the direct connection between two genes i and j and the neighboring genes that are shared between i and j. Each gene in the network is scored according to its overall interconnectivity to all starting nodes. Similarly to the network propagation method, the interconnectivity corrects for the node degree removing the potential bias introduced by the network topology. The interconnectivity between two nodes i and j is computed as follows:$$ICN\left(i,j\right)=\frac{2\cdot e\left(i,j\right)+n(i,j)}{\sqrt[{2}]{\deg\left(i\right)\cdot deg(j)}}$$

e(i,j) equals 1 if an edge between nodes i and j exists, and is set to 0 otherwise. n(i,j) corresponds to the number of neighboring nodes that are shared by i and j. deg(i) describes the degree of node i, i.e., the total number of its neighbors [42]. The final score for each network node is computed based on its overall interconnectivity to the starting points:$$Score\left(i\right)=\frac{1}{\left|Start\right|}\cdot \sum_{j\in Start}ICN(i,j)$$

Neighborhood scoring is a local method that only takes the direct neighborhood of the disease biomarkers into account [43]. In neighborhood scoring, each starting point is assigned a differential expression level corresponding to the fold change of the differentially expressed gene (1 in the current study case). This differential expression level is then smoothed over the gene’s direct neighborhood using the following formula:$${FC}_{i}^{\prime }=\alpha \cdot {FC}_{i}+\left(1-\alpha \right)\cdot \frac{\sum {FC}_{j}}{N}$$

For a gene i, its score (FC’_i_) is computed based on its own differential expression level (if applicable; this is the first part of the formula) and based on the average differential expression level of its neighbors j. α is a weighting factor and is set to 0.5 in this study.

### 4.4. Disease Similarity Approach

Starting from the contention that diseases with shared aspects of molecular pathology may serve as candidates for drug repositioning between them, disease similarity was investigated to predict further targets. To define disease similarity, the overlap between biomarkers was assessed by obtaining a metric for disease similarity as −log10(P) from a hypergeometric test. Disease similarity was used to select the 15 diseases most similar to the MCL disorders as a group. At least 3 targets had to be shared between the similar disease and the gold-standard training set. These similar disorders/symptoms were psychosis, memory impairment, sleep disorder, vascular dementia, panic disorder, alcoholism, spinal cord injury, mild cognitive impairment, renal failure, stroke, traumatic brain injury, atherosclerosis, ischemic stroke, obstructive sleep apnea and neurodegeneration. Next, each human protein received a disease-specific score of either 1 or 0 for being a target or not being a target, respectively.

### 4.5. Integration of the Evidence

The complementary methods yielded a vast array of input for drug target prediction. For optimal results, information was integrated in a weighted fashion by using machine learning (with higher weights assigned for more informative and smaller weights assigned for less informative methods). Weights were obtained by training the model on the already known targets, followed by application of the learned rule to prioritize all proteins as targets for MCL disorders. For training, the positive controls were defined as the targets associated with at least 2 indications from the predefined list of MCL disorders yielding the training set. The machine learning model was built using a partial least squares (PLS) regression algorithm. The predict function from the caret package was used to train PLS classifiers [44]. The analysis was performed using a balanced (stratified) nested cross-validation. The inner loop of the cross-validation feature selection using the recursive feature elimination approach was implemented. In the outer loop, the performance of the resulting classifiers that were trained on the best performing optimized feature set in the inner loop were tested. Both the outer and the inner loops represented 5-fold cross-validations. The negative controls were defined as a random subset of all targets that were not positive controls. This allowed for avoiding a strong imbalance between the number of positive and negative controls during the learning process. The negative controls were defined in each fold of the outer loop independently, so the effect of random selection was averaged out across the cross-validation. The number of negative controls was set to 10 times the number of positive controls. The entire cross-validation—from splitting the samples in outer folds and to integrative protein scoring—was repeated 10 times. Target PLS scores were converted into genome-wide ranks within each repeat. The ranks were finally averaged across the repeats to produce the final target ranking. Additionally, recursive feature elimination was performed within the inner loop of the cross-validation and model performance was assessed. Accordingly, change in prediction error with respect to the number of features used to build the model was evaluated by eliminating the least informative features one by one. The optimal number of features was selected for the final model, e.g., the model that corresponded to the minimal prediction error. Feature selection was performed 50 times: 10 repeats multiplied by 5 outer folds. For each feature, the number of times it was selected into the final model (a value from the 0–50 interval) was recorded. This quantity characterized the relative importance of each feature in the modeling process (see also the Appendix A below).

The performance of the final model was assessed using a receiver operating characteristics plot by plotting the true positive rate against the false positive rate for recovering the known control targets in the final target prioritization. The area under curve was used to characterize performance. Furthermore, to characterize model performance, the variation of the performance was assessed for the outer loop of the cross-validation across 10 repeats.

## 5. Conclusions

Our results provide molecular evidence for various mental disorders (e.g., Alzheimer’s disease, depression, anorexia nervosa etc.) possibly forming a coherent group of disorders with alterations of the mesocorticolimbic circuit. As MCL disorders seem to share substantial molecular aspects, therapeutic exploitation of novel targets may pave the way to new approaches to combat mental disorders and NCDs.

## Figures and Tables

**Figure 1 ijms-25-09682-f001:**
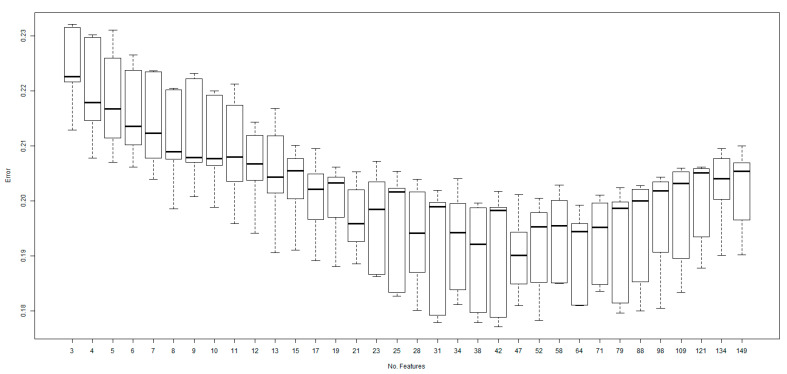
Predictive model performance characterized by prediction error as a function of the number of features included in the model. Features were removed successively; at each step the lowest ranked feature was removed. The lowest prediction error was seen at 47 features; thus, the final model was developed using the top 47 features. Initially (moving from right to left), prediction error falls as low impact features are removed. However, prediction error increases parallel to removing informative features.

**Figure 2 ijms-25-09682-f002:**
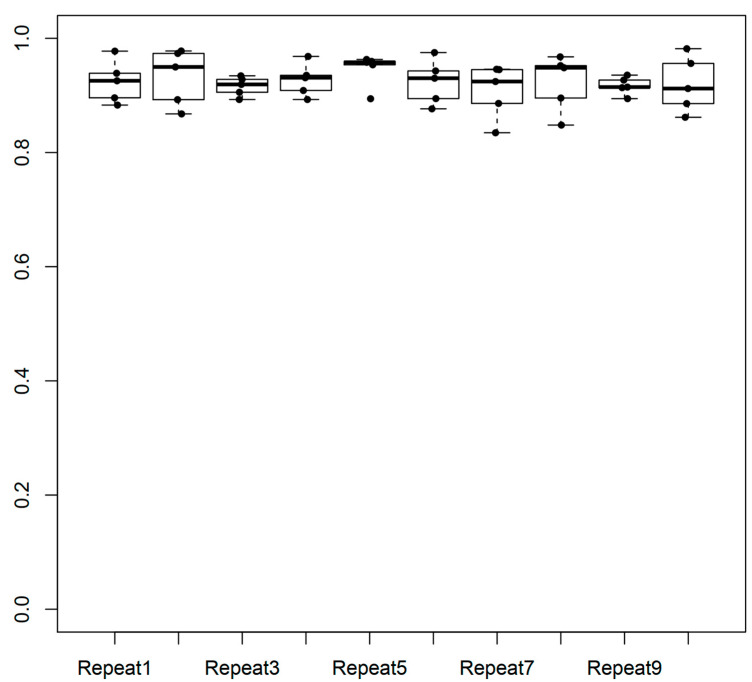
Variation of the performance of the predictive model across 10 repeats in the outer loop of the cross-validation of the training set (e.g., targets associated with at least 2 individual MCL disorders were included). Box plots correspond to distributions across the 10 analysis repeats.

**Figure 3 ijms-25-09682-f003:**
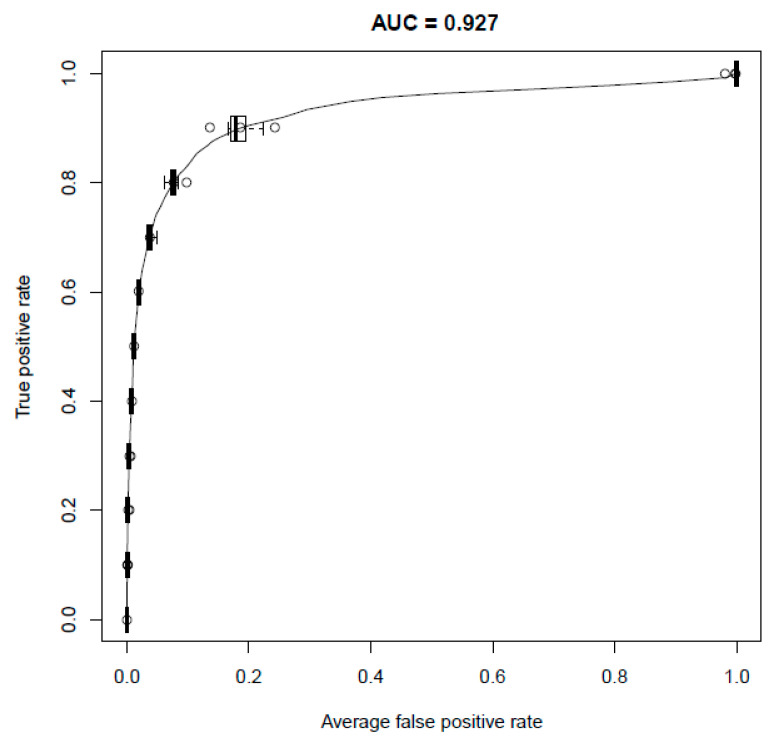
ROC curve of the training set (e.g., targets associated with at least 2 individual MCL disorders were included). Box plots correspond to curve distributions across the 10 analysis repeats.

**Figure 4 ijms-25-09682-f004:**
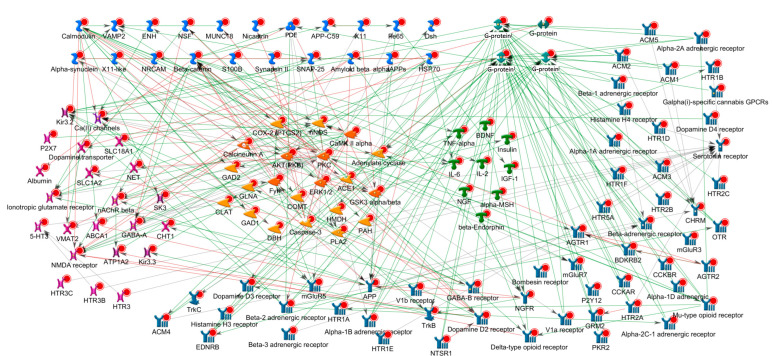
Literature-derived direct physical interactions connecting proteins from the top 250 target set. In this analysis, for each pair of proteins prior knowledge of direct interaction included in MetaBase^TM^ was assessed. Proteins with previous reports of direct interactions were connected with an arrow on the plot. Green arrows correspond to activating interaction mode, red—inhibition, grey—effect unspecified. Proteins are clustered by function: green (in the center)—ligands, yellow (in the center)—enzymes (kinases, phosphatases, etc.), blue (bottom, right)—plasma membrane receptors, purple (left)—ion channels, the four greenish proteins at the top are G-proteins; light blue at the top—proteins with miscellaneous functions. Red circles indicate that a protein belongs to the top 250 target list (all proteins on this plot). The top targets represent a dense interconnected system rather than a set of isolated molecules working independently in the cell.

**Table 1 ijms-25-09682-t001:** Top 25 control targets predicted for MCL disorders by machine learning. (Entrez IDs are available at https://www.ncbi.nlm.nih.gov/gene (accessed on 3 September 2024), gene symbols are indicated in accordance with the HUGO Gene Nomenclature Committee).

Entrez ID	Gene Symbol	Gene Name	Score
1813	DRD2	dopamine receptor D2	4.2
4129	MAOB	monoamine oxidase B	7.3
6531	SLC6A3	solute carrier family 6 (neurotransmitter transporter), member 3	10.5
4128	MAOA	monoamine oxidase A	16.0
3358	HTR2C	5-hydroxytryptamine (serotonin) receptor 2C, G protein-coupled	18.7
3351	HTR1B	5-hydroxytryptamine (serotonin) receptor 1B, G protein-coupled	19.6
1814	DRD3	dopamine receptor D3	20.4
6530	SLC6A2	solute carrier family 6 (neurotransmitter transporter), member 2	21.5
5970	RELA	v-rel avian reticuloendotheliosis viral oncogene homolog A	25.0
1137	CHRNA4	cholinergic receptor, nicotinic, alpha 4 (neuronal)	26.1
9177	HTR3B	5-hydroxytryptamine (serotonin) receptor 3B, ionotropic	32.6
320	APBA1	amyloid beta (A4) precursor protein-binding, family A, member 1	33.1
2904	GRIN2B	glutamate receptor, ionotropic, N-methyl D-aspartate 2B	39.3
4803	NGF	nerve growth factor (beta polypeptide)	40.5
3352	HTR1D	5-hydroxytryptamine (serotonin) receptor 1D, G protein-coupled	42.8
1815	DRD4	dopamine receptor D4	44.4
321	APBA2	amyloid beta (A4) precursor protein-binding, family A, member 2	46.1
2915	GRM5	glutamate receptor, metabotropic 5	46.5
3630	INS	insulin	47.7
3350	HTR1A	5-hydroxytryptamine (serotonin) receptor 1A, G protein-coupled	49.1
3354	HTR1E	5-hydroxytryptamine (serotonin) receptor 1E, G protein-coupled	50.6
3357	HTR2B	5-hydroxytryptamine (serotonin) receptor 2B, G protein-coupled	60.7
170572	HTR3C	5-hydroxytryptamine (serotonin) receptor 3C, ionotropic	73.7
2906	GRIN2D	glutamate receptor, ionotropic, N-methyl D-aspartate 2D	78.4
2668	GDNF	glial cell derived neurotrophic factor	91.2

**Table 2 ijms-25-09682-t002:** Enriched molecular functions of the top 25 known targets. r—number of genes with a given molecular function found within the top 25 known targets. R—number of analyzed targets (top 25). n—total number of genes associated with the function in the human genome. N—gene universe size (=number of genes associated with at least one known molecular function in the Gene Ontology database). *p*-value—enrichment *p*-value (hypergeometric test).

Molecular Function	r	R	n	N	*p* Value
serotonin binding	7	25	11	15,288	4.08 × 10^−18^
amine binding	7	25	13	15,288	2.12 × 10^−17^
drug binding	11	25	130	15,288	4.4 × 10^−17^
serotonin receptor activity	7	25	15	15,288	7.93 × 10^−17^
G-protein coupled amine receptor activity	7	25	42	15,288	3.23 × 10^−13^
transmembrane signaling receptor activity	14	25	1196	15,288	5.86 × 10^−10^
dopamine binding	4	25	10	15,288	1.16 × 10^−9^
signaling receptor activity	14	25	1299	15,288	1.74 × 10^−9^
signal transducer activity	15	25	1617	15,288	2.55 × 10^−9^
molecular transducer activity	15	25	1617	15,288	2.55 × 10^−9^
dopamine neurotransmitter receptor activity, coupled via Gi/Go	3	25	3	15,288	3.86 × 10^−9^
catecholamine binding	4	25	14	15,288	5.5 × 10^−9^
G-protein coupled receptor activity	11	25	812	15,288	1.99 × 10^−8^
receptor activity	14	25	1583	15,288	2.28 × 10^−8^
dopamine neurotransmitter receptor activity	3	25	5	15,288	3.85 × 10^−8^
extracellular ligand-gated ion channel activity	5	25	74	15,288	1.14 × 10^−7^
excitatory extracellular ligand-gated ion channel activity	4	25	49	15,288	1.12 × 10^−6^
ligand-gated channel activity	5	25	145	15,288	3.27 × 10^−6^
ligand-gated ion channel activity	5	25	145	15,288	3.27 × 10^−6^
neurotransmitter binding	3	25	24	15,288	7.64 × 10^−6^

**Table 3 ijms-25-09682-t003:** The top 25 novel targets predicted for MCL disorders via the machine learning approach. (Entrez IDs are available at https://www.ncbi.nlm.nih.gov/gene (accessed on 3 September 2024), gene symbols are indicated in accordance with the HUGO Gene Nomenclature Committee).

Entrez ID	Gene Symbol	Gene Name	Score
1803	DPP4	dipeptidyl-peptidase 4	14.8
3359	HTR3A	5-hydroxytryptamine (serotonin) receptor 3A, ionotropic	17.8
5468	PPARG	peroxisome proliferator-activated receptor gamma	29.5
1385	CREB1	cAMP responsive element binding protein 1	45.7
775	CACNA1C	calcium channel, voltage-dependent, L type, alpha 1C subunit	48.1
5443	POMC	proopiomelanocortin	51.9
2905	GRIN2C	glutamate receptor, ionotropic, N-methyl D-aspartate 2C	58.2
2903	GRIN2A	glutamate receptor, ionotropic, N-methyl D-aspartate 2A	61.9
6616	SNAP25	synaptosomal-associated protein, 25 kDa	65.5
9900	SV2A	synaptic vesicle glycoprotein 2A	69.6
776	CACNA1D	calcium channel, voltage-dependent, L type, alpha 1D subunit	73.2
781	CACNA2D1	calcium channel, voltage-dependent, alpha 2/delta subunit 1	73.9
783	CACNB2	calcium channel, voltage-dependent, beta 2 subunit	74.1
4842	NOS1	nitric oxide synthase 1 (neuronal)	81.5
1636	ACE	angiotensin I converting enzyme	82.1
801	CALM1	calmodulin 1 (phosphorylase kinase, delta)	83.0
784	CACNB3	calcium channel, voltage-dependent, beta 3 subunit	95.1
322	APBB1	amyloid beta (A4) precursor protein-binding, family B, member 1 (Fe65)	97.4
59285	CACNG6	calcium channel, voltage-dependent, gamma subunit 6	97.4
9254	CACNA2D2	calcium channel, voltage-dependent, alpha 2/delta subunit 2	98.3
19	ABCA1	ATP-binding cassette, sub-family A (ABC1), member 1	98.9
6285	S100B	S100 calcium binding protein B	100.7
185	AGTR1	angiotensin II receptor, type 1	101.7
773	CACNA1A	calcium channel, voltage-dependent, P/Q type, alpha 1A subunit	107.0
6570	SLC18A1	solute carrier family 18 (vesicular monoamine transporter), member 1	107.7

**Table 4 ijms-25-09682-t004:** Enriched molecular functions of the top 25 novel targets. r—number of genes with a given molecular function found within the top 25 novel targets. R—number of analyzed targets (top 25). n—total number of genes associated with the function in the human genome. N—gene universe size (=number of genes associated with at least one known molecular function in the Gene Ontology database). *p*-value—enrichment *p*-value (hypergeometric test).

Molecular Function	r	R	n	N	*p*-Value
voltage-gated cation channel activity	11	25	138	15,288	8.66 × 10^−17^
voltage-gated calcium channel activity	8	25	35	15,288	3.35 × 10^−16^
voltage-gated ion channel activity	11	25	188	15,288	2.78 × 10^−15^
voltage-gated channel activity	11	25	188	15,288	2.78 × 10^−15^
cation channel activity	12	25	289	15,288	6.93 × 10^−15^
ion gated channel activity	12	25	323	15,288	2.63 × 10^−14^
gated channel activity	12	25	323	15,288	2.63 × 10^−14^
calcium channel activity	9	25	109	15,288	6.32 × 10^−14^
calcium ion transmembrane transporter activity	9	25	128	15,288	2.77 × 10^−13^
ion channel activity	12	25	415	15,288	5.17 × 10^−13^
substrate-specific channel activity	12	25	425	15,288	6.85 × 10^−13^
high voltage-gated calcium channel activity	5	25	9	15,288	9.58 × 10^−13^
passive transmembrane transporter activity	12	25	450	15,288	1.34 × 10^−12^
channel activity	12	25	450	15,288	1.34 × 10^−12^
divalent inorganic cation transmembrane transporter activity	9	25	155	15,288	1.59 × 10^−12^
transmembrane transporter activity	15	25	972	15,288	1.81 × 10^−12^
cation transmembrane transporter activity	13	25	618	15,288	2.26 × 10^−12^
ion transmembrane transporter activity	14	25	825	15,288	4.08 × 10^−12^
metal ion transmembrane transporter activity	11	25	401	15,288	1.13 × 10^−11^

## Data Availability

The data presented in this study are available on request from the corresponding author. Furthermore, see the Appendix A.

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
