# Peer review of "Mesocorticolimbic and Cardiometabolic Diseases—Two Faces of the Same Coin?"

_ijms, 2024, doi:10.3390/ijms25179682_

Round 1
Reviewer 1 Report
Comments and Suggestions for Authors
Abstract missing implications to practice.
Sdgs need to be referred to UN not WHO.
Introduction lacked discussion on the current gaps in research that this study aims to address. Its only until discussion authors explain that this study aimed to identify common targets for mesocorticolimbic system disorders as a group and test a hypothesis that these disorders may be approached therapeutically as a group. This argument and why need to be in intro. Personalized medicine is key.
There is also gap with hypothesis “to test the hypothesis that these pathologies may be approached therapeutically as a group and whether novel therapeutic targets may be identified”. As I mentioned a comment in abstract authors need to explicitly mention the potential clinical benefits of identifying novel therapeutic targets.
"MetaBase™" and "IntegritySM" are need to be defined for non-technical people.
Thomson Reuters databases use is ok but more detail on the specific algorithms and parameters used in the machine learning process would enhance reproducibility. Authors need to add codes of analysis as supplemental.
Authors stated 137 predefined positive control genes/targets were used. Please provide more details on how these control targets were selected.
Tables are difficult to read due to lack of notes. E.g. in table 1 authors need to reference the Entrez ID and targets are standardized according to a recognized nomenclature such as HGNC (HUGO Gene Nomenclature Committee). And if Control target all yes why this column is needed. Known Associations need to be added.
Fig 1/AUC is not enough additional metrics such as precision, recall, and F1 score should be included to provide a more comprehensive evaluation of the model's performance.
Results reported AUC 0.927/93% which indicates good model performance. However, a more detailed discussion on the potential reasons for this high accuracy and how it compares to previous studies would be beneficial. Did authors try to enhance precision?
Better discussion and review of the mechanisms linking MCL dysfunction with NCDs and CMDs is needed.
The limitations of the study need to be discussed.
Authors need to add next steps in this line of research particularly the experimental validation of the identified targets and the potential for clinical trials.
English need revision: The risk behaviors underlying the most prevalent chronic noncommunicable diseases (NCDs) include alcohol misuse, unhealthy diet, smoking and sedentary lifestyle, behaviors. Some English check is needed: “The risk behaviors associated with the most prevalent chronic noncommunicable diseases (NCDs) encompass alcohol misuse, unhealthy diet, smoking, and a sedentary lifestyle.
Abstract missing implications to practice.
Sdgs need to be referred to UN not WHO.
Introduction lacked discussion on the current gaps in research that this study aims to address. Its only until discussion authors explain that this study aimed to identify common targets for mesocorticolimbic system disorders as a group and test a hypothesis that these disorders may be approached therapeutically as a group. This argument and why need to be in intro. Personalized medicine is key.
There is also gap with hypothesis “to test the hypothesis that these pathologies may be approached therapeutically as a group and whether novel therapeutic targets may be identified”. As I mentioned a comment in abstract authors need to explicitly mention the potential clinical benefits of identifying novel therapeutic targets.
"MetaBase™" and "IntegritySM" are need to be defined for non-technical people.
Thomson Reuters databases use is ok but more detail on the specific algorithms and parameters used in the machine learning process would enhance reproducibility. Authors need to add codes of analysis as supplemental.
Authors stated 137 predefined positive control genes/targets were used. Please provide more details on how these control targets were selected.
Tables are difficult to read due to lack of notes. E.g. in table 1 authors need to reference the Entrez ID and targets are standardized according to a recognized nomenclature such as HGNC (HUGO Gene Nomenclature Committee). And if Control target all yes why this column is needed. Known Associations need to be added.
Fig 1/AUC is not enough additional metrics such as precision, recall, and F1 score should be included to provide a more comprehensive evaluation of the model's performance.
Results reported AUC 0.927/93% which indicates good model performance. However, a more detailed discussion on the potential reasons for this high accuracy and how it compares to previous studies would be beneficial. Did authors try to enhance precision?
Better discussion and review of the mechanisms linking MCL dysfunction with NCDs and CMDs is needed.
The limitations of the study need to be discussed.
Authors need to add next steps in this line of research particularly the experimental validation of the identified targets and the potential for clinical trials.
English need revision: The risk behaviors underlying the most prevalent chronic noncommunicable diseases (NCDs) include alcohol misuse, unhealthy diet, smoking and sedentary lifestyle, behaviors. Some English check is needed: “The risk behaviors associated with the most prevalent chronic noncommunicable diseases (NCDs) encompass alcohol misuse, unhealthy diet, smoking, and a sedentary lifestyle.
Author Response
Please find our responses, attached

Reviewer 2 Report
Comments and Suggestions for Authors
In the current study, the authors verified the interesting idea of incorporating common molecular pathways in the course of non-communicable diseases and common mental disorders into new therapeutic strategies for both disorders. Machine learning was used to identify potential factors. In my opinion, the results presented in this work will be valuable for guiding future research in this field.
Line 44 - please use the reference format used by MDPI
Line 68 - the hypothesis is misconstrued, at the moment I do not see the element of scientific hypothesis there. Rather, it is a research objective.
Line 178 - the abbreviation PONC is undefined. On the other hand, the abbreviation PLSR on page 414 is not used once in the text. Please check if the other abbreviations have expansions and the validity of their introduction.
Author Response
Please find our responses, attached

Reviewer 3 Report
Comments and Suggestions for Authors
The manuscript is extremely interesting, I have few observations on the methodology.
1- Please confirm that the IntegritySM website (http://integrity.thomson-pharma.com) is correct, and whether its access is in the public domain or not.
2- As for MetaBase™ , also describe the form of access (website, etc) and whether it is in the public domain.
Author Response
Please find our responses, enclosed

Round 2
Reviewer 1 Report
Comments and Suggestions for Authors
thank you for addressing concerns.